# Interplay of valley, layer and band topology towards interacting quantum phases in moiré bilayer graphene

Yungi Jeong [1,2], Hangyeol Park[1,2], Taeho Kim [2], Kenji Watanabe [3], Takashi Taniguchi [4], Jeil Jung [5,6] & Joonho Jang [1,2] ✉

In Bernal-stacked bilayer graphene (BBG), the Landau levels give rise to an intimate connection between valley and layer degrees of freedom. Adding a moiré superlattice potential enriches the BBG physics with the formation of topological minibands – potentially leading to tunable exotic quantum transport. Here, we present magnetotransport measurements of a high-quality bilayer graphene–hexagonal boron nitride (hBN) heterostructure. The zero-degree alignment generates a strong moiré superlattice potential for the electrons in BBG and the resulting Landau fan diagram of longitudinal and Hall resistance displays a Hofstadter butterfly pattern with a high level of detail. We demonstrate that the intricate relationship between valley and layer degrees of freedom controls the topology of moiré-induced bands, significantly influencing the energetics of interacting quantum phases in the BBG superlattice. We further observe signatures of field-induced correlated insulators, helical edge states and clear quantizations of interaction-driven topological quantum phases, such as symmetry broken Chern insulators.

Bernal bilayer graphene (BBG) is the simplest member of the rhombohedral stacked multilayer graphene family. By breaking the inversion symmetry of BBG with a perpendicular electric displacement field $D$, BBG shows a tunable gap opening and van Hove singularities which can lead to a cascade of symmetry broken states[1–4]. Also, electrons in BBG have a layer degree of freedom correlated with the valley degree of freedom in a specific way, which makes it possible to investigate the proximity effect of substrates surrounding BBG. When BBG is aligned with hexagonal boron nitride (hBN), the slight lattice mismatch (~1.8%) of the layers creates a periodic moiré superlattice potential. Especially in this system, the zero-degree rotation between BBG and hBN is known to be the most stable with the global configurational energy minimum; thus it is expected to have very low superlattice

disorder. This superlattice potential is expected to modify the band structure near the K and K' points and induces secondary Dirac points and flat bands with non-trivial topological properties[5]. Remarkably, in the presence of a perpendicular magnetic field, the so-called Hofstadter's butterfly appears due to the interplay between superlattice potential and magnetic field[6–9]. When the magnetic flux per superlattice unit cell becomes a rational multiple of the flux quantum, a commensurability between the cyclotron orbits and the lattice periodicity leads to regain a lattice translational symmetry to form extended electronic states known as Brown–Zak (BZ) quasiparticles[10–13]. These BZ quasiparticles, satisfying the new Bloch equation, move through the lattice as if the magnetic field is zero. Away from these commensurable magnetic field values, BZ quasiparticle

[1]Center for Correlated Electron Systems, Institute for Basic Science, Seoul 08826, Korea. [2]Department of Physics and Astronomy, and Institute of Applied Physics, Seoul National University, Seoul 08826, Korea. [3]Research Center for Electronic and Optical Materials, National Institute for Materials Science, 1-1 Namiki, Tsukuba 305-0044, Japan. [4]Research Center for Materials Nanoarchitectonics, National Institute for Materials Science, 1-1 Namiki, Tsukuba 305-0044, Japan. [5]Department of Physics, University of Seoul, Seoul, Korea. [6]Department of Smart Cities, University of Seoul, Seoul, Korea. ✉e-mail: joonho.jang@snu.ac.kr

feels an effective magnetic field $B_{eff} = B - B_{p/q}$ (penetration of p flux quanta per q superlattice unit cells.) and form mini Landau fans radiating from $B = B_{p/q}$ lines. Because BZ quasiparticles can be defined for each rational $p/q$, a fractal-like self-repeating structure - called the Hofstadter butterfly[14–19] – would emerge.

In addition to single-particle effects, electron–electron interactions can play a significant role in these systems due to the relatively flat moiré bands, resulting in a large density of states at certain energies. These interactions lead to quantum correlated phases, and their interplay with the band topology adds another layer of controllability and richness to the physics of BBG/hBN systems. By tuning the $D$ field, which affects the interlayer potential, one expects to control the topological properties and electrical energy bandwidth of the system, offering a pathway to explore a range of interacting topological quantum states and their phase transitions, and potentially realize devices where the topological property is dynamically controlled. Understanding of the interplay between the valley and layer physics of BBG and the superlattice potential from hBN thus is critical. However, despite the desirable combination of the stable configuration and electric- and tunability, the electron transport properties in a BBG/hBN moiré superlattice are relatively less explored compared to the other moiré systems, mostly due to the technical difficulty in making high-quality samples with both top- and bottom-gate electrodes and robust ohmic transport contacts necessary in a strong magnetic field and superlattice potential. In this article, we report a magneto transport study of a high mobility zero-degree aligned BBG/hBN heterostructure with dual graphite gates and four high-transparent contacts that allow successful longitudinal and Hall measurements. The high quality of the sample and the zero-degree alignment allow us to observe a large variety of Chern insulator states and interaction-driven states even under relatively low magnetic fields, revealing an intricate interplay of various quantum degrees of freedom.

## Results

### Device characterization

BBG/hBN aligned sample was fabricated by a conventional dry-transfer method using PDMS and PC film[20] (Fig. 1a). Figure 1b shows the stacking structure of the dual-gated BBG/hBN aligned device. Since conventional metal contacts can open gaps in the BBG[21,22], an additional graphite layer was used to ensure that contact resistance remains sufficiently low even when strong magnetic and displacement fields are applied in cryogenic temperatures. We performed four-probe electrical transport measurements of the sample. Figure 1c shows the resistance as a function of carrier density at zero magnetic and displacement field. Due to the superlattice potential imposed by the lattice mismatch between hBN and BBG, induced energy gap or DOS minimum is expected at the energies below and above the charge neutral point (CNP)[7,8]. Actually, a recent experiment supports DOS minimum rather than full gap[23] (see also Supplementary Note 9). We indeed observe two satellite peaks on either side of CNP at positive and negative densities, $n = \pm 2.397 \times 10^{12}$ cm$^{-2}$ ($= \pm 4n_0$), where the superlattice-induced isolated energy bands are either completely full or empty. From the value, we estimate that the lattice constant of the superlattice is 13.88 nm and determine that the twist angle between the BBG and the top hBN is 0 degree within our experimental accuracy. Figure 1d shows the Hall mobility as a function of carrier density, and the mobility near the CNP is about 200,000 cm$^2$ V s$^{-1}$ comparable to high mobility GaAs[24–26] or suspended graphene[27,28], indicating that the sample is of very high quality. Unlike the free-standing BBG, we notice that the mobility dips near the full filling the isolated superlattice

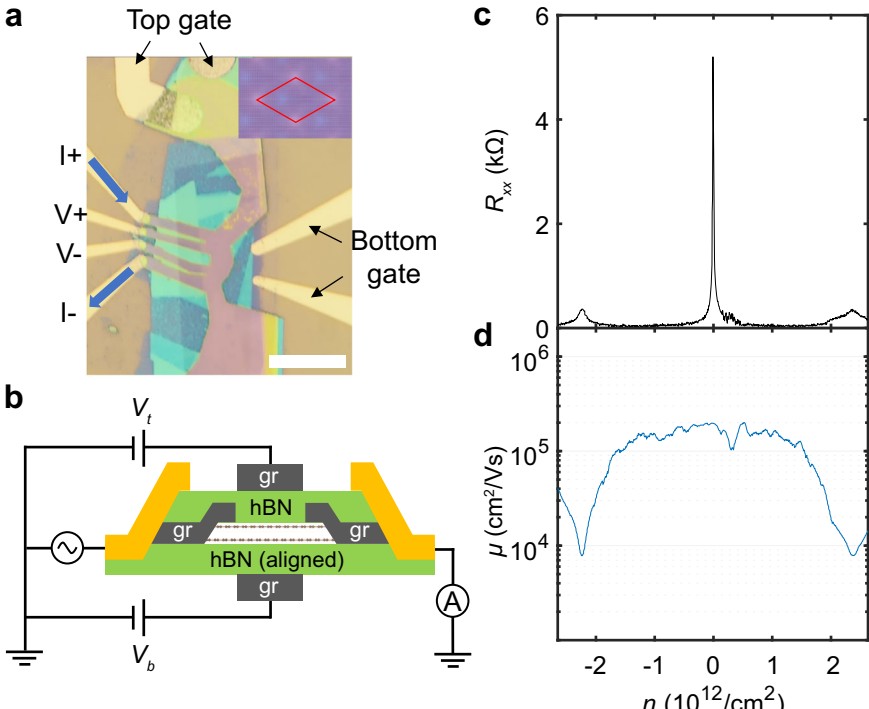

**Fig. 1 | Device geometry and zero magnetic field transport measurements.**
**a** Optical microscope image of the aligned BBG/hBN heterostructure device. Scale bar, 15 μm. The crystalline axis of bottom hBN and BBG was aligned at almost 0 degrees. Inset: Schematic of the moiré pattern seen in the 0 degree aligned BBG/hBN device. **b** Schematic diagram of the BBG/hBN aligned device. gr denotes the graphite flake. Top and bottom gates make it possible to tune the carrier density and $D$ field simultaneously. And the graphite contact layer was additionally inserted to make good ohmic contact to the BBG. **c** Longitudinal resistance versus carrier density at zero magnetic and displacement field. Satellite peaks were observed at $n = \pm 2.397 \times 10^{12}$ cm$^{-2}$ on either side of the CNP peak. **d** Hall mobility versus carrier density at zero magnetic and displacement field. Mobility decreases as the carrier density approaches the superlattice full fillings from CNP.

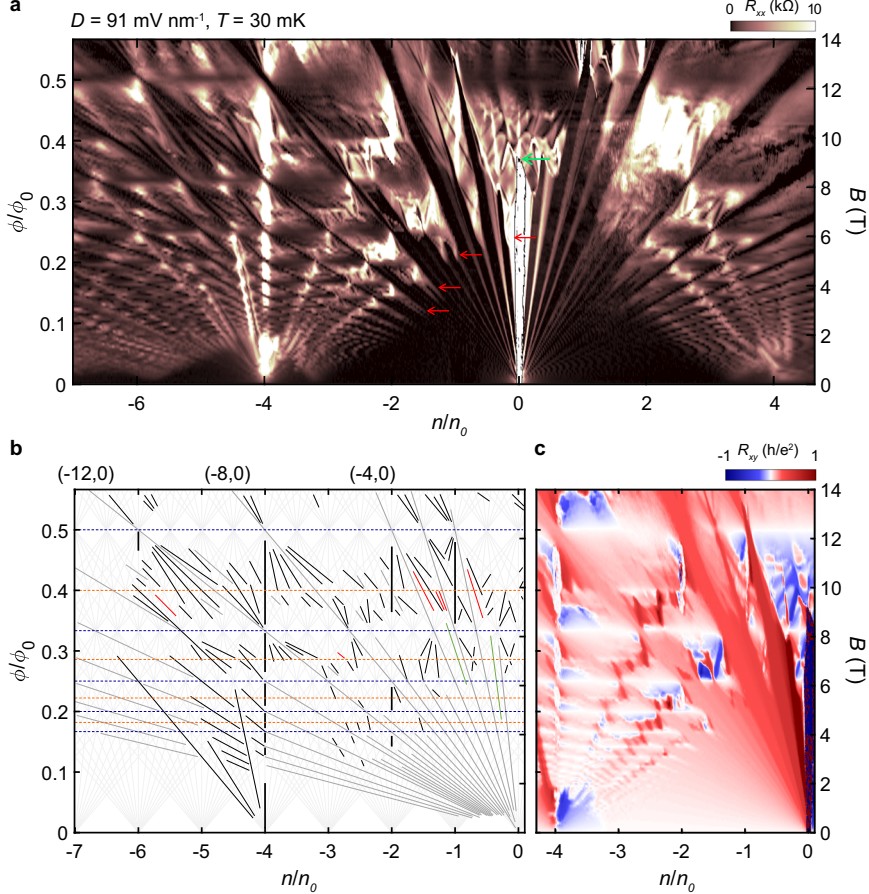

**Fig. 2 | Magneto transport in a BBG/hBN moiré superlattice. a** Longitudinal resistance, as a function of normalized carrier density and magnetic flux, measured at $T = 30$ mK with $D = 91$ mV nm$^{-1}$. Here, n is the carrier density, $n_0$ is the density corresponding to full filling of the superlattice, $\phi$ is the magnetic flux per super-lattice unit cell and $\phi_0 = h/e$ is the magnetic flux quantum. The red arrows point to some Landau levels that are strongly influenced by the moiré potential and exhibit high resistance. The green arrow indicates the point where the bandgap at CNP closes under the effect of moiré potential. The color scale is truncated at 10 kΩ. **b** A Wannier diagram to denote observed incompressible states identified in **a**. We assume that both spin and valley degeneracies are lifted, and only show up to $|t| \le 8$.

Light gray lines indicate incompressible states allowed by Diophantine equation assuming spin and valley degeneracy are lifted. Four classes of trajectories are distinguished by color: Integer quantum Hall insulators (gray; integer $t$, $s = 0$), Fractional Quantum Hall insulators (green; fractional $t$, $s = 0$), moiré potential induced Chern insulators (black; integer $t$, integer $s$, $s \neq 0$), Symmetry-broken Chern insulators (red; integer $t$, fractional $s$). The blue and orange horizontal lines represent the first and second order Brown–Zak oscillations ($\phi/\phi_0 = 1/q, 2/q$), respectively. **c** Hall resistance data in the hole-doped region at the same temperature and $D$ field.

bands, probably due to the small number of effective charge carriers near the full filling of moiré band.

## Magneto transport measurement

Figure 2 shows longitudinal and Hall resistance measurements in the presence of a perpendicular magnetic field (see also Supplementary Fig. 1, 2 for various $D$ field values). In Fig. 2a, the dark regions of low longitudinal resistance are identified as incompressible states with edge states. The effects of the superlattice in the spectrum are readily noticeable; at the intersections between the horizontal lines at $\phi/\phi_0 = p/q$ and lines of the conventional integer quantum Hall states, mini fans appear to form the fractal-like Hofstadter's butterfly. In the Hall resistance measurements in Fig. 2c, we directly identify the effective magnetic field $B_{eff} = B - B_{p/q}$ felt by BZ quasiparticles with sign changes of Hall resistance at $\phi/\phi_0 = p/q$, forming a horizontal pattern of white strips. We also find the $D$ field-induced insulating gap at CNP abruptly closes near $B = 9$ T due to the superlattice effects (green arrow in Fig. 2a; see also Supplementary Fig. 3). In the low magnetic field region near CNP, the Landau fan looks qualitatively similar to the case of an intrinsic BBG with the spin–valley subbands of Landau levels resolved even below B = 1 T but, in higher magnetic fields, the splittings

become less distinct and even disappear, due to the dominant effect of the moiré potential leading to significant overlaps between LL sub-bands, while ushering in the appearance of moiré-induced incompressible states.

Each incompressible state in the Landau fan spectrum follows the Diophantine equation $n/n_0 = t(\phi/\phi_0) + s$, where the slope $t$ gives the total Chern number of occupied bands proportional to the Hall conductance, and the $n$-intercept $s$ gives the number of charges trapped in the superlattice unit cell. Unlike the conventional quantum Hall states ($s = 0$), nonzero $s$ states have charge density and Hall conductance decoupled by a strong superlattice potential[19,29]. To investigate these states thoroughly, we plotted a Wannier diagram in Fig. 2b, where the light gray lines represent all possible trajectories allowed by the Diophantine equation. Then, we overlaid additional colored lines that correspond to the observed insulators in Fig. 2a. First, the dark gray lines correspond to the conventional integer quantum Hall effect (IQHE) (integers $t$ and $s = 0$), and the black lines identify the Chern insulators (CI) (integer $t$, integer $s \neq 0$); all these states can appear in a non-interacting single-particle Hofstadter spectrum. On the other hand, due to particle interaction, more incompressible features appear in the data especially when

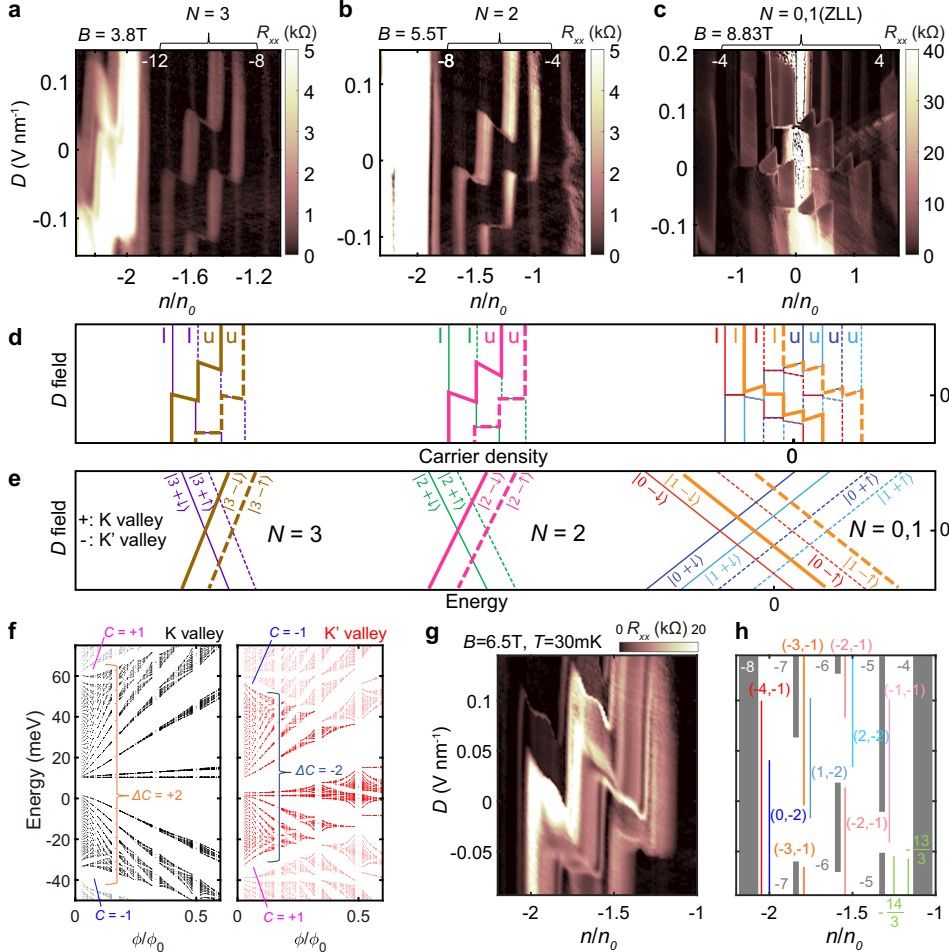

**Fig. 3 | Degeneracy lifted Landau levels with moiré superlattice potential.**
**a**–**c** Longitudinal resistance $R_{xx}$ as a function of carrier density and $D$ field at $B$ = (**a**) 3.8 T, (**b**) 5.5 T, and (**c**) 8.23 T ($\phi/\phi_0$ = 1/3) at $T$ = 30 mK. $N$ denotes the orbital index of the Landau level (LL) of BBG, and white-colored numbers indicate the Landau filling factors ν of integer quantum Hall states. Integer quantum Hall states appear dark due to its dissipationless edge state, while metallic states that partially fill the LL have relatively high resistance and appear bright. **d** Schematic of filling sequences in **a**–**c**. l (lower) and u (upper) denote the layer polarization of each state. **e** Schematic of LL energy spectrum. Each state is denoted by $|N\xi\sigma\rangle$ (orbital, valley, and spin). The tendency of the energy with respect to the $D$ field is derived from a single-particle model, and the filling sequence of zero-energy LL ($N$ = 0,1) measured at $T$ = 30 mK. For example, the green lines indicate the

additionally takes into account Coulomb interactions[50] (see also Supplementary Note 7). In **d**–**e** colors indicate the type of $N\xi$, with solid lines for spin down and dashed lines for spin up. And, the bold lines indicate levels that are strongly influenced by the moiré potential in **a**–**c**. **f** moiré-induced electronic energy spectra calculated for each valley. Note the Chern number $\Delta C$ of the moiré-induced iso-lated bands is different for each valley, leading to different valley-dependent broadening behaviors. **g** $D$ field tunability of Chern insulator states in $N$ = 2 Landau levels. **h** Schematic of incompressible states in **g**. Each gap is color-coded and labeled ($t$,$s$). Only $t$ values are shown for IQHE and FQHE. Each color scale is truncated at the end value of its respective colorbar.

fractional quantum Hall effect (FQHE) (fractional $t$, $s$ = 0), where the well-known FQHE at $\nu = m - 1/3$ ($m = \pm 1, \pm 3$)[30] are visible as low as $B$ = 5 T, reflecting the high quality of the electron system. And several red lines denote symmetry broken Chern insulator (SBCI) (integer $t$, fractional $s$) whose trajectories are located at fractional filling of moiré Chern bands. We emphasize that most of the Chern numbers of the incompressible states are independently confirmed by the Hall quantization measurements.

### $D$ field tunable valley-selective moiré effect

The electronic system displays dramatic changes in spectra when tuned with $D$ field, due to BBG's layer degrees of freedom and its peculiar property associated with the valley degrees of freedom. In particular, striking bright fork-like features appear and strongly depend on $D$ field (red arrows in Fig. 2a, Supplementary Fig. 1). These fork-like features, when plotted as a function of $D$ and $n$ as in Fig. 3a–c, appear as thick bright lines. Interestingly, for higher Landau levels (LLs) with $N \geq 2$ (Fig. 3a–b), the bright lines move in the opposite

direction to the case for the zero-energy Landau levels (ZLLs) (Fig. 3c, Supplementary Fig.4). This phenomenon is depicted in the schematic of Fig. 3d. To explain this observation, we consider the energies of the LLs of BBG given by[31],

$$
\begin{cases}
\epsilon_0 = \frac{1}{2}\xi U + E_s\sigma \\
\epsilon_1 = \left(\frac{1}{2} - \frac{\hbar\omega_c}{\gamma_1}\right)\xi U + E_s\sigma + \Delta_{10} \\
\epsilon_N^\pm = \pm\hbar\omega_c\sqrt{N(N-1)} - \frac{\hbar\omega_c}{2\gamma_1}\xi U + E_s\sigma \, (N \geq 2)
\end{cases}
\tag{1}
$$

Here, $N$ is the orbital index of Landau levels, $\omega_c$ is the cyclotron frequency of carriers in BBG, $\xi$ is the valley index (+1 for K valley, −1 for K' valley), $\gamma_1$ is the interlayer hopping parameter[32,33], $U$ is the interlayer potential given as $U = Dd_0$ ($d_0$: interlayer distance of BBG), $E_s$ is the spin-dependent energy splitting, that includes the Zeeman energy, to lift spin degeneracy of the bands, and $\Delta_{10} \propto \gamma_4 B$ ($\gamma_4$ is the skew tunneling term from non-dimer carbon site to dimer carbon site[32,33]) is an additional energy difference between $N$ = 0 and 1.

Because $\hbar\omega_c/\gamma_1$ is smaller than 1/2 for the range of magnetic fields used in our experiment, $\epsilon_0$, $\epsilon_1$ for ZLLs ($N = 0,1$) and $\epsilon_N$ for higher LLs ($N \geq 2$) have the opposite coefficients of energy shifts upon the change of $U$ (and thus $D$) for a specific valley; i.e. for a fixed valley, the layer polarization $p = -\partial\epsilon/\partial D$ of a higher LL has the opposite sign to the one of a ZLL. Thus, the energy of the K (K') valley should increase (decrease) with increasing $D$ field in ZLLs, while it decreases (increase) in higher LLs. The schematic in Fig. 3e based on this valley analysis fully explains the behavior of the bright features in Fig. 3a–c. It thus strongly suggests that the two out of the four symmetry-broken subbands of a LL selectively experience the stronger superlattice effect and that all the bright bands are actually of a specific valley (denoted as K' valley for both ZLLs and LLs with $N \geq 2$) experiencing more scattering that results in a higher resistance. We point out that this finding is somewhat counterintuitive and contrary to the previous beliefs[7,8] in that the degrees of freedom that determine the effect of the moiré potential are not the layers but actually the valleys (Fig. 3d, e). We further performed numerical simulation (Supplementary Fig. 15) that supports the phenomenology of significant subband broadenings for one of the valleys. Interestingly, the topology of the superlattice-induced isolated bands play an important role here; the combined Chern number of the induced isolated bands has the exact opposite value for each valley as shown in Fig. 3f (see also Supplementary Fig. 16), and we find that the observed valley-selective moiré effect originates from the disparate magnetic field responses of the isolated bands with opposite Chern numbers because the band edges merge at lower magnetic fields for the isolated band (of K' valley) with negative Chern number, inducing more severe modification of LL spectra[5,34,35]. More discussions on the potential microscopic mechanism of this phenomenon are in Supplementary Note 6.

The gate tunability further extends to the topological properties of the moiré bands. By applying $D$ field, we control the valley-selective effect of moiré potential, indirectly via the layer polarization, whose values are peculiarly intertwined with the valleys. In particular, various Chern insulator states are controlled due to the interplay of valleys and layers upon varying the vertical $D$ field. In Fig. 3g, h, we plotted data showing the $D$ field tunability of the $N = 2$ LL. Inside each LL subbands, multiple Chern gaps appear and disappear strongly depending on $D$ field values. We find that the Chern insulators transit under a certain rule: while IQHE gap at $\nu = 5, 6, 7$ closes with LL subbands switching their positions in filling sequence, the Chern insulators whose Chern numbers $t$ differ by 1 (or −1) but with the same $s$ values appear in the adjacent LL subbands in a cascading fashion (Supplementary Fig. 5)

This suggests the Chern insulators are valley- and spin-polarized, and adds to the idea that tuning valley or spin degrees of freedom of the Chern insulators is a key to control topology in this BBG system.

## Correlated insulating states

Another outstanding feature in the spectra is the existence of the insulating states located at $n/n_0 = -1, -2$ surprisingly persistent throughout the values of $B$, as shown in Fig. 4. At $B = 0$, there is an insulating phase at $n/n_0 = 0$, but upon increasing the magnetic field, new insulators develop at $n/n_0 = -1, -2$ (blue arrows in Fig. 4a). We evaluated the bulk insulating gap of $(0,-2)$ state by fitting the temperature dependence to the Arrhenius formula (Fig. 4b), and found the gap size is particularly larger than other nearby superlattice-induced Chern insulating states. This is not explainable by our simulation based only on the single-particle picture (Supplementary Fig. 14), and thus strongly suggests that electron correlation strongly enhances the energy gaps of the states. Such a correlated insulator can emerge when particle interaction leads to spontaneous spin or valley order and saves energy by filling the superlattice with one particle per unit cell[36]. We attribute this strong correlation to the narrow bandwidth of the isolated valence band between the CNP (at $n = 0$) and superlattice-induced insulator at $n/n_0 = -4$. According to the simulation (Supplementary Fig. 13), the bandwidth is to be smaller than 50 meV and decreases upon increasing the strength of $D$. Certainly, most of the fractional states and correlated insulating states we have observed exist between $n/n_0 = 0$ and $-4$, suggestive of the narrow moiré-induced isolated band playing an important role in enhancing the interaction effects. In addition, we changed the configuration of the contact as in Supplementary Fig. 8b to measure the non-local resistance ($R_{NL}$) of the sample and found that $R_{NL}$ is significantly large at $(0,-1)$. We attribute the high non-local resistance of the state to a helical edge channel of counter-propagating valleys (see Supplementary Note 9). However, we cannot rule out the complicated involvement of spins in the edge channel, such as one in a spin−valley magnetic state. A measurement with a large in-plane magnetic field to control Zeeman energy may help to resolve the question while it is out of the scope of the current work.

## Fractional incompressible states

The effect of particle interaction due to the narrow bandwidth (Fig. 4) and tunable band topology (Fig. 3) further leads to exotic incompressible fractional states. Remarkably, in Fig. 5a–c and Supplementary Fig. 7, we clearly identify multiple phases whose values of $s$ are

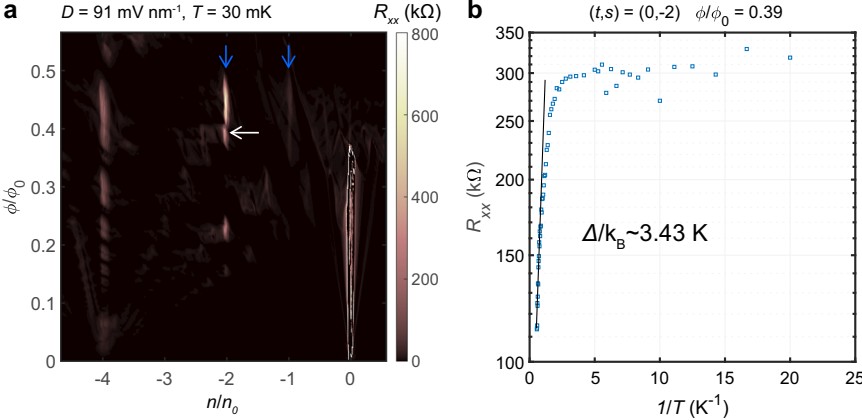

**Fig. 4 | Correlated insulating states. a** Longitudinal resistance, as a function of normalized carrier density and magnetic flux, measured at $T = 30$ mK with $D = 91$ mV nm⁻¹. Blue arrows indicate insulating states at $n/n_0 = -1, -2$ respectively. The color scale is adjusted to avoid saturations at $t = 0$ features. **b** Temperature

dependence of longitudinal resistance of $(0,-2)$ state at $\phi/\phi_0 = 0.39$ (indicated by a white arrow in **a**). The gap was estimated by fitting the Arrhenius formula $R_{xx} = R_0 \exp(\Delta/2k_B T)$.

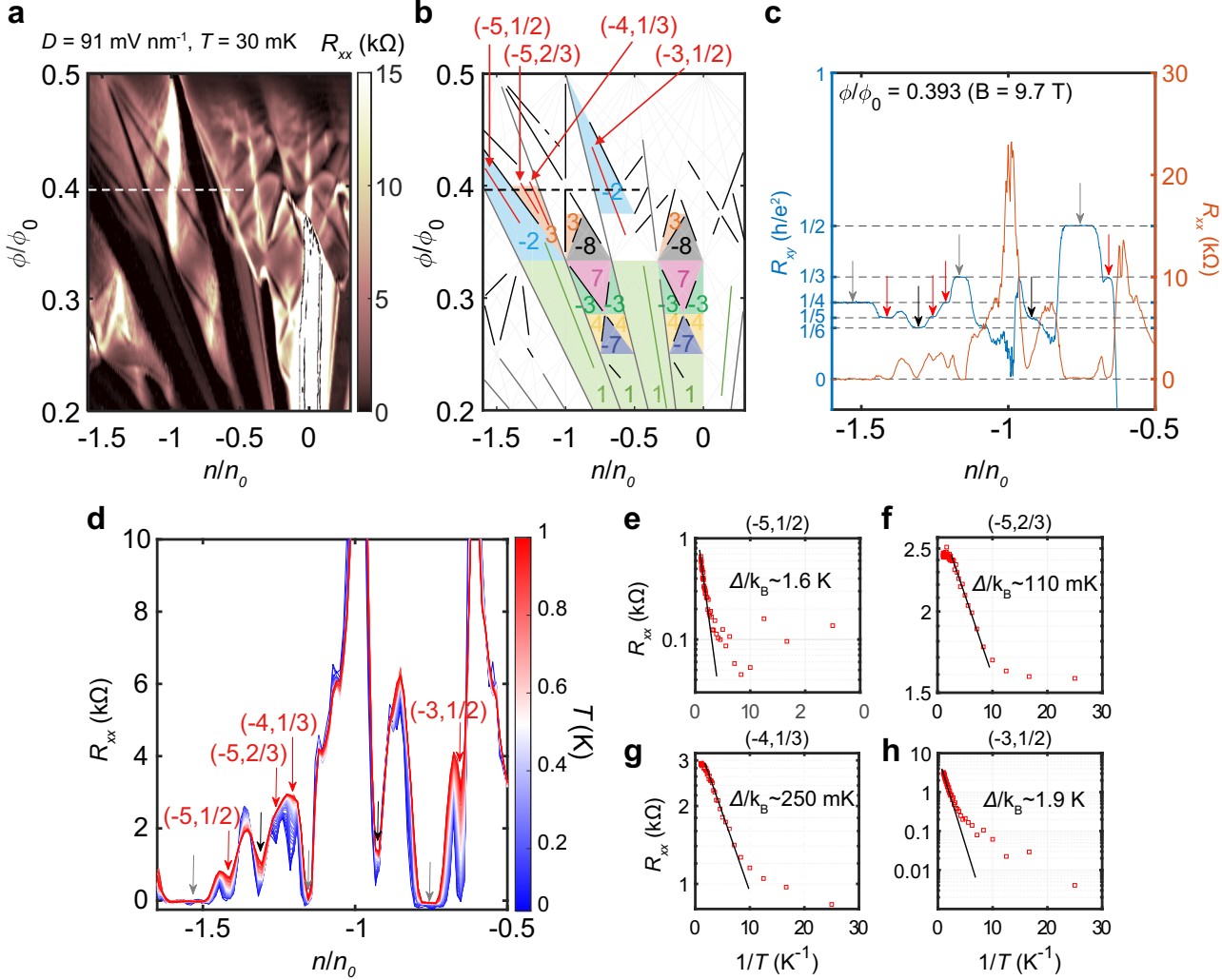

**Fig. 5 | Interaction-driven fractional states. a** A zoomed-in area of Fig. 2a (see also Supplementary Fig. 1f). The color scale is truncated at 15 kΩ. **b** The Wannier diagram to denote observed states in **a**. Each incompressible state line is labeled with the same color as in Fig. 2b, and each colored region represents the single-particle Chern bands with Chern numbers $\Delta t$ obtained by evaluating the $t$-difference between the neighboring single-particle Chern gaps. The patterns of the Chern bands are similar for each of the $N = 0$ (first and third from left) and $N = 1$ (second and fourth) ZLLs. **c** Line cuts along the $\phi/\phi_0 = 0.393$ line (dashed horizontal line in **a** and **b**). The blue graph on the left axis shows the Hall resistance in units of $h/e^2$ with dashed lines representing $1/t$. The orange graph on the right axis shows the longitudinal resistance. Each incompressible state is marked with the same colored arrow as in **b** and shows quantized plateau in Hall resistance and dip in longitudinal resistance. **d** Temperature dependence of longitudinal resistance along the $\phi/\phi_0 = 0.389$ line. Each of the incompressible states exhibited thermal activation behavior, with resistance tending to increase with increasing temperature (see also Supplementary Fig. 6). **e–h** Temperature dependent longitudinal resistance of fractional states denoted in **d**. Gaps were estimated by fitting the Arrhenius formula with edge state $R_{xx} = R_0 \exp(-\Delta/2k_BT)$.

fractional even below B = 7 T, which is 2–3 times lower than previous reports[29,37,38] and another manifestation of the high quality of this sample. These observed interaction-driven states have gap trajectories described by $(t,s) = (t_L, s_L) + \nu_C(\Delta t = t_R - t_L, \Delta s = s_R - s_L)$ (where $(t_{L,R}, s_{L,R})$ is the neighboring left and right integer states respectively) with a fractional $\nu_C$. We observed states with $\nu_C = 1/2$ in bands with Chern number $\Delta t = -2$ ((−5,1/2), (−3,1/2)) and states with $\nu_C = 1/3$ and 2/3 in a band with $\Delta t = 3$ ((−5,2/3), (−4,1/3)). These (integer $t$, fractional $s$) states are interpreted as symmetry broken Chern insulators (SBCIs) arising from the spontaneous doubling/tripling of the superlattice unit cell because they do not follow the theoretically expected filling factor $\nu_C$ of fractional Chern insulators (FCIs) in a Chern band with $|\Delta t| > 1$[39,40]. However, we still cannot exclude the possibility of partial filling of the moiré Chern band with exotic fractionalized excitations. The techniques that can detect charge quantization, such as shot noise measurements[41], would be helpful to resolve this issue. In Fig. 5e–h, the energy gaps of the SBCI states were extracted by fitting to the

Arrhenius formula. We can divide these four states into two relatively large (Fig. 5e, h) and two relatively small (Fig. 5f, g) gap pairs. These two pairs have different Chern numbers of the fractal bands supporting each state, which suggests that the band topology is closely related to the nature of the fractional states (see also Supplementary Fig. 7f, g). Also, we observed that unlike the conventional FQH states in a misaligned BBG appearing in a relatively wide range of $D$ field[30,42,43], fractional states in our sample exist only in a narrow range of $D$ field and seem to critically depend on the fractal band's bandwidth and topology, defined by the band Chern number.

## Discussion

In conclusion, we have shown the field-dependent interacting Hofstadter spectrum in BBG/hBN moiré system and discovered that, upon application of $D$ field, the system's valleys respond according to their LL-dependent layer polarization and generate distinct spectral features that strongly depend on the valley-dependent band Chern

number. Also, we experimentally identified interaction-driven states in the fractal bands based on their definitive fractional Hall quantization. Our work demonstrates that owing to the in-situ tunability with gate electrodes and the high-quality assisted by its mechanical stability, BBG/hBN moiré system provides a highly-tunable platform for studying the interplay of band topology and electron correlation, and opens up exciting opportunities to explore spin–valley isospin polarizations of interacting states in Hofstadter spectrum. Further theoretical investigations on the energetics of SBCIs and FCIs and its quantitative relationship to the Chern number of fractal bands would be highly desired.

## Methods

### Device fabrication

The encapsulated hBN (aligned)/BBG/hBN device was fabricated using the van der Waals dry-transfer technique. The entire stack was picked up by PDMS/PC stamp in the following order: hBN (for graphite pickup), top graphite gate, top hBN (34 nm), contact graphite, BBG, bottom hBN (61 nm), bottom graphite gate. Graphite flakes were used only after ensuring that they had at least 7–8 layers by optical contrast. The stack was dropped down on the pre-defined align marker pattern and annealed in a vacuum furnace 500 degrees for 2 h to accumulate small bubbles in the stack. After electron beam lithography, aluminum was deposited and used as an etch mask, and since the contact graphite is only in contact with one side of the BBG, the sample is etched in the shape of a horseshoe rather than a Hall bar. Gate and contact graphite were edge contacted with Ti/Au metallic leads. Finally, just before measurement, the sample was annealed in a vacuum furnace at 400 degrees for 1 h.

### Measurement

Measurements were taken in a cryogen-free dilution refrigerator with a base temperature of 20 mK. Due to the slight temperature increase caused by operating the superconducting magnet, most of the actual measurements were performed at 30 mK. An RC/RF electric filter and a sapphire stripline heat sink[44] were used to lower the electron temperature. Electrical measurements were performed using standard lock-in amplifier techniques. To measure longitudinal resistance, an AC voltage bias of $1\,mV_{RMS}$, 13.33 Hz was connected to the top contact of the device in series with a 1 MΩ resistor to simulate an AC current bias of $1\,nA_{RMS}$. The lowest contact was then connected to the current input of the SR865A lock-in amplifier to measure the AC current (Fig.1a). Voltage was measured between the second and third contacts in the middle of the device using a SR560 voltage preamplifier with a gain of 1000. Then the longitudinal resistance was defined as $R_{xx} = V/I$. Due to the large parameter space (carrier density, displacement field, magnetic field, temperature), most of the measurements were performed by one-shot measurement or by taking a small number of measurements and averaging them to reduce the measurement time. However, if the parameter sweep speed is not set appropriately, it can cause measurement errors, so it is important to set the appropriate saturation time according to the measurement frequency and the time constant of the amplifier.

### Numerical simulation

We performed a numerical simulation of the electronic energy spectra in a BBG/hBN with zero-degree alignment by the direct diagonalization of a Hamiltonian matrix following the continuum model proposed by Bistritzer and Macdonald[45]. Due to the natural lattice mismatch of 1.08%, electrons in the BBG experience a superlattice potential imposed at the interface to the hBN even when the twisted angle is zero, which is known to be the energetically stable configuration after considering the lattice relaxation. At a finite perpendicular magnetic field, the following Hamiltonian becomes numerically solvable when expressed in the bases of Landau

gauge eigenfunctions (usually truncated at LLs above $N \sim 200$ for the manageable computation time, but still with a good approximation).

$$\bar{H} = \begin{pmatrix} (-U-\Delta_{sub})/2 & -\pi & v_4\pi & v_3\pi^\dagger & 0 & 0 \\ -\pi^\dagger & (-U+\Delta_{sub})/2 & \gamma_1 & v_4\pi & 0 & 0 \\ v_4\pi^\dagger & \gamma_1 & (U-\Delta_{sub})/2 & -\pi & & T \\ v_3\pi & v_4\pi^\dagger & -\pi^\dagger & (U+\Delta_{sub})/2 & & \\ 0 & 0 & & T^\dagger & U_B & 0 \\ 0 & 0 & & & 0 & U_N \end{pmatrix} \quad (2)$$
$$(\text{where } \pi \equiv \xi p_x - ip_y, \; p_i \equiv -i\hbar\nabla_i - eA_i)$$

(where $\pi \equiv \xi p_x - ip_y$, $p_i \equiv -i\hbar\nabla_i - eA_i$), where it is written in the bases of $|\psi\rangle = (|A1\rangle, |B1\rangle, |A2\rangle, |B2\rangle, |\bar{B}\rangle, |\bar{N}\rangle)$. Here, $|A1\rangle$ and $|B1\rangle$ are the bases of the top graphene sublattices, $|A2\rangle$ and $|B2\rangle$ are of the bottom graphene, and $|\bar{B}\rangle$ and $|\bar{N}\rangle$ are the atomic Boron and Nitrogen subbases of h$\bar{B}\bar{N}$, respectively. Then, we have used the following parameters in the simulation: $v = 9.1\times10^7$ cm/s, $\gamma_1 = 400$ meV, $v_3 = 9.0\times10^6$ cm/s, $v_4 = 4.5\times10^6$ cm/s, $U_B = -1400$ meV and $U_N = +3300$ meV[7,46,47]. We set $\Delta_{sub} = 0$ meV. The matrix $T$ represents the interlayer moiré hopping between the bottom graphene and hBN and is written as,

$$T = \left[ \begin{pmatrix} u & u' \\ u' & u \end{pmatrix} + e^{i\xi\mathbf{G}_1'\cdot\mathbf{r}} \begin{pmatrix} u & u'\omega^{-\xi} \\ u'\omega^\xi & u \end{pmatrix} + e^{i\xi(\mathbf{G}_1'+\mathbf{G}_2')\cdot\mathbf{r}} \begin{pmatrix} u & u'\omega^\xi \\ u'\omega^{-\xi} & u \end{pmatrix} \right] \quad (3)$$

where we used two different parameters $u' = 130$ meV, $u = 0.8\times130$ meV to account for the lattice relaxation effect[48,49], and $\omega = e^{2\pi i/3}$. $\mathbf{G}$'s are the reciprocal lattice vectors of the lattice mismatch-induced moiré potential. Note the four blocks boxed with dotted-lines hybridize two monolayer graphene blocks and a hBN block, so that the Hamiltonian represents the whole BBG/hBN heterostructure. By setting the blocks labeled as $T$ and $T^\dagger$ to 2-by-2 null matrices, one recovers the intrinsic BBG spectra.

## Data availability

The source data used in this study are available in the figshare database under accession code https://doi.org/10.6084/m9.figshare.24119286. Other data that support the findings of this study are available from the corresponding author upon request.

## Code availability

The codes related to the findings of this study are available from the corresponding authors upon request.

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

## Acknowledgements

We gratefully acknowledge helpful discussions with B. J. Yang. The work at SNU was supported by the National Research Foundation of Korea grants funded by the Ministry of Science and ICT (Grant Nos. 2019R1C1C1006520, 2020R1A5A1016518, RS-2023-00258359), the Institute for Basic Science of Korea (Grant No. IBS-R009-D1), SNU Core Center for Physical Property Measurements at Extreme Physical Conditions (Grant No. 2021R1A6C101B418), Creative-Pioneering Researcher Program through Seoul National University and Samsung DS Basic Research Program (Project No. 0409-20230298). J. J. acknowledges support from Samsung Science and Technology Foundation Grant No. SSTF-BA1802-06. K. W. and T. T. acknowledge support from the JSPS KAKENHI (Grant Numbers 21H05233 and 23H02052) and World Premier International Research Center Initiative (WPI), MEXT, Japan.

## Author contributions

Y.J. and J.Jang conceived the project, Y.J. fabricated the device and performed measurements with the help from H.P. and T.K. Y.J. and J.Jang analyzed data and performed numerical calculations with the help from J.Jung. K.W. and T.T. grew the single crystal hBN. Y.J., J.Jung and J.Jang wrote the manuscript with inputs from all authors. J.Jang supervised the overall project.

## Competing interests

The authors declare no competing interests.
