## [Peer Review File · Nature Communications]

REVIEWER COMMENTS

Reviewer #1 (Remarks to the Author):

In this work, the authors perform magnetotransport measurements in a moire superlattice created by aligning Bernal-stacked bilayer graphene with hexagonal boron nitride. The measured Hofstadter butterfly spectrum shows a high level of detail, with first and second order Brown-Zak oscillations visible, and multiple incompressible states corresponding to integer quantum Hall effect, Chern bands, as well as ones beyond the single-particle picture as fractional quantum Hall states, symmetry broken Chern insulators, and fractional Chern insulators. Further, they demonstrate the tunability of Chern insulator states by the displacement field in the samples, and additionally find insulating states manifesting themselves as vertical features at filling factors $-1, -2,$ and -4 . The data presented for the fractional t or s phases show a remarkable fractional quantization of the Hall resistance. The results are promising for a dynamical control of topological quantum states. Overall, the work studies in detail a variety of phenomena, and in general the conclusions are well justified. The results are significant, and in my opinion, suitable for publishing in Nature Communications. Some minor revisions can be made for clarity, as listed below.

- In the caption of Fig. 3, Lines 170-171, as well as in the corresponding main text, the authors present the states filling sequence as a function of displacement field, based on the energy spectrum of the Landau levels derived from the single-particle model. For the zero Landau level, additionally the Coulomb interactions are taken into account, according to a sentence in the caption. However, I failed to find information in the text or supplementary file on how exactly the Coulomb interactions were included in this model.

- (From line 237) Regarding the insulating states manifesting as vertical features at fillings $n/n_0 = -1, -2, -4$, the authors suggest that the possible origin is beyond the single-particle model. While the gap at half filling $(t,s) = (0, -2)$ hints at an origin due to electron correlation, similarly as in Ref. [34], could the authors also explain in more detail the nature $(0, -1)$ state? Moreover, in Ref. [34] the insulating state at half filling is suppressed by an external magnetic field of the order of 6 T, due to the Zeeman effect. The transport data presented in this work show a gap appearing at comparably higher magnetic fields -- the gap is still visible at about 12 T. The curious behavior seen in this work deserves a comment.

- In the paragraph starting in line 274, the authors identify phases with fractional t or s . Since the FCI are states with fractional both t and s , could the authors clarify their statement that the state $(t,s) = (-4, 1/3)$ can also be classified as an FCI?

Reviewer #2 (Remarks to the Author):

The authors present a transport study of Bernal bilayer graphene, aligned to BN. These systems feature a rich Hofstadter butterfly spectrum, with many competing states, building on the initial observation of (Dean, Nature 2013). Through extensive measurements of $R_{xx}(T,n,D)$ and $R_{xy}(T,n,D)$, the authors isolate and analyze integer and fractional Chern insulating states in one device. The clean and detailed measurements will be of interest to specialists in the field, together with the authors' exploration of the valley degree of freedom as driver for symmetry-broken states.

However, I cannot recommend the manuscript as it is for publication until the authors address the following points:

-Figure 4 and its discussion relies on a two-terminal (2T) measurement. The details on how this was calculated should be included in the Methods section. If I assume the authors define the 2T resistance as $(1V/I_{rms} - 1M\Omega)$, where I_{rms} is the current measured by the SR865, this means the contributions of the graphite/BBG interface, the graphite leads, the metallic leads etc. are still present. This makes the attribution of R_{2T} to physics in the BBG/BN doubtful. The resistance of the graphite leads will depend on the applied gate voltage, and its temperature dependence will also contribute to $R_{2T}(T)$. Four-terminal measurements are the standard precisely to exclude these effects. Can the authors provide four-terminal data to eliminate contributions from other sources?

-The entire manuscript rests on a single device. If at all possible, can the authors present data from a similar device that support the conclusions? At zero degree BBG/BN alignment, making another device does not rely on precise angle control. High-quality stacks should "snap" back to zero degrees alignment upon annealing.

-The thickness of each layer should be mentioned in the Device Fabrication section. Especially the thickness of the BN spacers between BBG and the gates is important, as the screening by the gates can influence the existence of the fragile Chern insulating states.

-The authors mention that "conventional metal contacts can open gaps in BBG" (line 92). A citation for this would be appreciated, and a value of the contact resistance of the graphite contacts at 30 mK and high magnetic field.

-Figure 1d shows the mobility of the device. Is this the Hall mobility, or the mobility from the onset of SdH oscillations? How relevant is the mobility of a device that is guaranteed to be ballistic at 30 mK in this discussion? The clear Hofstadter butterfly of Figure 2 is enough indication of the cleanliness of this device. In the ultra-quantum limit, the quantum lifetime τ_q is the only useful measure of the quality of a device.

Reviewer #3 (Remarks to the Author):

Summary of Presented Work

The authors present magnetotransport measurements on a Bernal bilayer graphene device with its crystal axes aligned to one of its encapsulating hexagonal boron nitrides. They observe a rich set of states associated with the moire pattern formed by alignment to the hBN, most of which can be explained by the known Landau Level structure of bilayer graphene and Hofstadter spectrum resulting from the moire pattern and high magnetic field. They analyze in more detail than previous works the effects of displacement field and the longitudinal resistance. Their main experimental claims are:

- 1) The observation of valley-dependent (rather than layer-dependent) moire-induced scattering observed as “bright” R_{xx} features in displacement field vs. carrier density plots.
- 2) Identification of quantum valley Hall insulators with insulators with $t=0$.
- 3) Observation of interaction-driven states, specifically FCI and SBCI observed for the first time in R_{xy} plateaux.

Major comments

Overall, this paper presents the most complete transport picture of hBN-aligned BBG so far. As a result, the cacophony of states make parsing this data quite a task even for experts. I believe that the work requires another pass to make it more digestible to a broader audience, including clarifying the visual presentation of figures, and perhaps presenting more closely zoomed in regions of the data in Fig.5 , for example.

- 1) The valley-dependent broadening is an intriguing observation and in my mind well supported by the data.
- 2) I am not convinced that one can confidently identify the $t=0$ features shown in Fig.4 as the quantum valley Hall effect. Claiming the existence of helical edge states based on only a non-monotonic temperature dependence is well below the standard applied to Quantum spin Hall

systems both in semiconductor heterostructures and more relevantly at $\nu = 0$ in monolayer graphene. As evidenced by the data itself, there are many competing interacting and non-interacting states in the phase diagram, so I am not sure that one can attribute the non-monotonic dependence to edge states vs. disordered competition between multiple states with different, temperature-dependent resistances.

For the claims of edge states, some sort of reconfiguration of contacts showing length-dependent transport should be possible to bolster these claims.

Additionally, it would be useful to explain in more detail why one would expect the QVH effect instead of quantum spin Hall, some more complex spin-valley magnetic state, or a charge density wave here would be useful in bolstering the claim of QVH.

3) The very nice R_{xy} data and gap measurements are very welcome additions, however I have concerns with the identification of the FCI state, in particular.

The authors identify a single FCI state with fractional t in their data at $(-14/3, 5/9)$, located in a $C=3$ band originating at $1/3$ of a flux quantum per moire unit cell. I do not believe the manuscript provides enough evidence that this state is not in fact another nearby SBCI state $(-5, -2/3)$. I'll outline why I think this is the case below:

a) The plateau at $R_{xy} = 3/14 h/e^2$ is not convincing. If the authors would like to make this claim, the plateau should be more clearly presented. The R_{xx} minimum of this state is the least developed of the R_{xx} dips, as one would expect from the very small measured gap. Strictly speaking, this means that the plateau should not be quantized., and based on what I can see could just as well be a poorly developed $1/5 h/e^2$ plateau. I think a better presentation and discussion of the plateau is warranted in either case.

b) In the absence of more clear data, I think that while an FCI is possible at these t and s values, it is much more likely that it is energetically favorable for the $C=3$ band to breaking into three, symmetry broken $C=1$ bands due to interactions, than an FCI appearing in an already symmetry broken $C=2$ band.

A back of the envelope way of estimating the strength of interactions in the Hofstadter spectrum is to look at the single-particle bands 'width' in carrier density at a given field, which we can take as the overall degeneracy of the band (or similarly we can relate it to an effective magnetic length which governs Coulomb interactions). The state in question appears while the triangle of the $C=3$

band is still quite narrow, much narrower than the LL at 5 Tesla in which the traditional FQH state first appears. So, for an FCI to appear in the regions indicated by Fig.5, it would have to somehow be more robust than the highest energy gap FQH state in the system, which seems very unlikely to me given the finite bandwidth associated with the Hofstadter butterfly. I think a positive identification of this state would require a lot of theoretical explanation beyond what is provided.

Summary

To sum up, I think that the data and much of the discussion here is very intriguing to experts in the subfield, however there are major outstanding issues I would need to see addressed to recommend publication on Nature Communications. Please also see some more minor comments and corrections provided below.

In terms of significance I believe this work is original and of interest to many people working on graphene and other 2D materials.

Minor comments:

Around line 98, it is stated that energy gaps are expected, but only very small peaks in R_{xx} are observed. A brief comment on this would be helpful to the reader.

Fig 2b is very busy, specifically it is very hard to distinguish the QH gray lines and the Diophantine gray lines. Personally, I also think that the Brown Zak lines could be removed or made less intrusive.

For all of the resistance figures, it should be stated or otherwise indicated on the colormap that the color scale is truncated at 10, 15, etc. kOhms, as there are much more resistive features washed out by the choice of color limits (necessarily).

Fig.3d,e. The thick dashed lines are confusing as the gaps between dashes are very thick and at first glance appeared to be marking places where features didn't exist.

Between Fig5b and C the colors of arrows for the different states perhaps should match.

I'd like to see any characterization of the graphite contacts, as from personal experience and discussions with others, I would expect graphite contacts to be uniquely bad for transport at high fields, especially if the crystal axis of the graphene and graphite are not very close.

We would like to thank the reviewers for the useful comments which helped us improve our manuscript. We did our best to answer the questions and comments raised by the reviewers, not only by providing further discussions but also by conducting additional experiments that support our claims. Following the suggestions by the reviewers, we added new experimental results to the main text and the supplementary information. A list of major changes in the revised manuscript is given below, followed by point-by-point responses to the reviewers' comments.

List of changes.

1. The two-probe measurement data in **Fig. 4** is replaced with four-probe measurement data where the graphite contact effect was removed.
2. Magnetoresistance measurement data from newly fabricated BBG-hBN aligned devices are added to the supplementary information.
3. The nature of vertical insulating features is further discussed. Data supporting the existence of helical edge states are newly added.
4. The phase we previously claimed to be FCI is now correctly identified as SBCI based on additional Hall resistance measurements.
5. Several figures are revised and added to improve readability.

Reply to reviewer 1's report

In this work, the authors perform magnetotransport measurements in a moire superlattice created by aligning Bernal-stacked bilayer graphene with hexagonal boron nitride. The measured Hofstadter butterfly spectrum shows a high level of detail, with first and second order Brown-Zak oscillations visible, and multiple incompressible states corresponding to integer quantum Hall effect, Chern bands, as well as ones beyond the single-particle picture as fractional quantum Hall states, symmetry broken Chern insulators, and fractional Chern insulators. Further, they demonstrate the tunability of Chern insulator states by the displacement field in the samples, and additionally find insulating states manifesting themselves as vertical features at filling factors -1,-2, and -4. The data presented for the fractional t or s phases show a remarkable fractional quantization of the Hall resistance. The results are promising for a dynamical control of topological quantum states. Overall, the work studies in detail a variety of phenomena, and in general the conclusions are well justified. The results are significant, and in my opinion, suitable for publishing in Nature Communications. Some minor revisions can be made for clarity, as listed below.

Author's response: We are very happy to hear the reviewer's positive opinions on our work. We also greatly appreciate the reviewer's helpful comments. We made every effort to address the points raised by the reviewer.

- In the caption of Fig. 3, Lines 170-171, as well as in the corresponding main text, the authors present the states filling sequence as a function of displacement field, based on the energy spectrum of the Landau levels derived from the single-particle model. For the zero Landau level, additionally the Coulomb interactions are taken into account, according to a sentence in the caption. However, I failed to find information in the text or supplementary file on how exactly the Coulomb interactions were included in this model.

Author's response: We appreciate the reviewer for raising an important point. Since we didn't explicitly explain how the Coulomb interactions were included in the filling sequence of ZLL, we have added the following paragraph to the supplementary information.

“If we draw the energy spectrum (or the filling sequence) of the zero Landau level (ZLL) based on the single particle model (1) in the main text, it is as shown in **Fig. R1a**, which is different from **Fig. 3e (Fig. R1b)**. To explain the difference, let us consider the large $D > 0$ case on the hole side (i.e., the upper left part of **Fig. R1 a, b**).

The single particle picture of ZLL in BBG with small Zeeman energy expects the filling sequence from $\nu = -4$ should be $|N\xi\sigma\rangle = |0-\downarrow\rangle, |0-\uparrow\rangle, |1-\downarrow\rangle, |1-\uparrow\rangle$ for large $D > 0$ case. However, Coulomb interactions change this filling order. Coulomb interaction favors sequential filling of $N=0,1$ with same spin due to the exchange interaction.¹ i.e. filling different orbitals with same spin, (e.g. $|0-\downarrow\rangle, |1-\downarrow\rangle$) has more favorable Coulomb energy than filling same orbital of opposite spin (e.g. $|0-\downarrow\rangle, |0-\uparrow\rangle$).

The experimental filling sequence (orbital, valley, spin) of ZLL in our sample was identified by analyzing the Landau fan diagram and n - D sweep. First, we analyzed

the orbital nature ($N=0$ or 1) of the hole side of ZLL at the large $D>0$ case (i.e. the upper left side of **Fig. 3c**). By observing the ZLL in **Fig. 2a** or **5a**, we can see that filling $-4<\nu<-3$ looks similar to $-2<\nu<-1$ (both shows conventional fractional quantum Hall state up to $\phi/\phi_0 = 1/3$), while $-3<\nu<-2$ looks similar to $-1<\nu<0$ (a similar moiré Chern band structure emerged from $\phi/\phi_0 \approx 0.2$). This similarity is due to the same orbital nature of each pair² and the single particle calculation in the ref.(2) showed that the moiré Chern band was developed at a lower magnetic field for $N=1$ state. This supports the filling order $|N\xi\sigma\rangle=|0-\downarrow\rangle, |1-\downarrow\rangle, |0-\uparrow\rangle, |1-\uparrow\rangle$ for large $D>0$ case in our sample (large D makes valley $\xi = -$ fills before valley $\xi = +$). And in the reference cited in the main text³, a detailed experimental and theoretical account of the ZLL in the absence of a moiré potential was provided. In the ref. (3), the filling sequence of ZLLs according to the size of the interaction is simulated, and it can be seen that in the weak interaction regime, the ZLLs are filled in the order of $N=0,0,1,1$ as discussed above, and in the intermediate and strong interaction regime, the ZLLs are filled in the order of $N=0,1,0,1$. By tracking two states feeling moiré potential strongly ($|N\xi\sigma\rangle=|1-\downarrow\rangle, |1-\uparrow\rangle$) in **Fig. 3c**, we can see that our data is more consistent with the intermediate interaction regime confirmed experimentally in ref. (3).

Additionally, if the magnetic field is further increased, the filling sequence mentioned above can be flipped once more. Since Δ_{10} is proportional to B , while the Coulomb interaction E_C is proportional to \sqrt{B} , there will be a point where $\Delta_{10} > E_C$ as the magnetic field increases. In this case, the single particle effect will dominate again, and the electrons will be filled in order $|0-\downarrow\rangle, |0-\uparrow\rangle, |1-\downarrow\rangle, |1-\uparrow\rangle$. However, in our experimental setup, we did not reach the magnetic field where this reversal occurs.”

Fig. R1 | Schematic of ZLL of BBG energy spectrum in purely single particle picture (a) and interaction considered (b). Note that b is purely illustrative since the actual energetics, taking into account interactions, do not simply come out as a straight line. Also, the specific energy gap and slope differences were ignored in b.

- (From line 237) Regarding the insulating states manifesting as vertical features at fillings $n/n_0 = -1, -2, -4$, the authors suggest that the possible origin is beyond the single-particle model. While the gap at half filling $(t,s) = (0,-2)$ hints at an origin due to electron correlation, similarly as in Ref. [34], could the authors also explain in more detail the nature $(0,-1)$ state? Moreover, in Ref. [34] the insulating state at half filling is suppressed by an external magnetic field of the order of 6 T, due to the Zeeman effect. The transport data presented in this work show a gap appearing at comparably higher magnetic fields -- the gap is still visible at about 12 T. The curious behavior seen in this work deserves a comment.

Author's response: We conducted additional non-local transport measurements to analyze the vertical features at fillings $n/n_0 = -1, -2$, and -4 and found that the $(0,-1)$ state is expected to have a helical edge state. (**Fig. R6**) Noting this, we deduced that $(0,-1)$ is likely to be a state having the valley-dependent Chern numbers with opposite signs at K and K', with suppressed intervalley scattering. But, the possibility of a spin Hall state or a complex spin-valley magnetic state cannot be ruled out, and we have added a discussion of this in the main text and supplementary.

For the second question, our picture is that the two systems (MATBG vs. aligned BBG-hBN) have different causes for the formation of the single-particle flat band. In the case of MATBG, the flat band is formed at zero field by interlayer hybridization, but in the case of aligned BBG-hBN, the bandwidth at zero field is on the order of tens of meV, which is not very large, but still about 5-10 times larger than that of MATBG. Therefore, it is expected to be difficult to observe correlated insulators at zero field. However, upon applying a magnetic field, the BBG-hBN system forms a moire induced Chern band at the single particle level, and if the bandwidth of the Chern band is sufficiently small, a correlated insulator can emerge by interaction.

- In the paragraph starting in line 274, the authors identify phases with fractional t or s . Since the FCI are states with fractional both t and s , could the authors clarify their statement that the state $(t,s) = (-4, 1/3)$ can also be classified as an FCI?

Author's response: We thank the reviewer for pointing out the potentially misleading expression. In the original manuscript, we thought that for interaction driven states with gap trajectory $(t,s) = (t_L, s_L) + \nu_C(\Delta t, \Delta s)$, if there is an exotic fractional excitation whose fractional filling factor ν_C is an integer multiple of $1/\Delta t$, resulting in (integer t , fractional s), then these states can also be thought of as a kind of fractional Chern insulator (FCI) in a broad sense. However, in general, the theoretically expected filling factor of FCIs in a Chern band with higher Chern number (i.e. $|\Delta t| > 1$) is of the form $\nu_C = \frac{r}{(2r\Delta t + 1)}$ (r is a nonzero integer)^{4,5}, so these FCIs always have fractional t values. Therefore, such FCIs and the exotic fractional excitations mentioned above should be distinguished in a strict sense.

Thus, we removed the potentially misleading statement, and modified the manuscript as follows.

In the original manuscript:

“The state with $(-4, 1/3)$ is a curious one, in that while it could be simply seen as a SBCCI, we cannot rule out the possibility of the state being another FCI based on the calculation $(-4, 1/3) = (-3, 0) - 1/3 \times (3, -1)$.”

In the revised manuscript:

“These (integer t , fractional s) states are interpreted as symmetry broken Chern insulators (SBCIs) arising from the spontaneous doubling/tripling of the superlattice unit cell because they do not follow the theoretically expected filling factor ν_C of fractional Chern insulators (FCIs) in a Chern band with $|\Delta t| > 1$. However, we still cannot exclude the possibility of partial filling of the moiré Chern band with exotic fractionalized excitations.”

Reply to reviewer 2's report

The authors present a transport study of Bernal bilayer graphene, aligned to BN. These systems feature a rich Hofstadter butterfly spectrum, with many competing states, building on the initial observation of (Dean, Nature 2013). Through extensive measurements of $R_{xx}(T,n,D)$ and $R_{xy}(T,n,D)$, the authors isolate and analyze integer and fractional Chern insulating states in one device. The clean and detailed measurements will be of interest to specialists in the field, together with the authors' exploration of the valley degree of freedom as driver for symmetry-broken states.

However, I cannot recommend the manuscript as it is for publication until the authors address the following points:

Author's response: We appreciate the reviewer for acknowledging the importance of our work. In the following we believe we have addressed the most points raised by the reviewer.

-Figure 4 and its discussion relies on a two-terminal (2T) measurement. The details on how this was calculated should be included in the Methods section. If I assume the authors define the 2T resistance as $(1V/I_{rms} - 1M\Omega)$, where I_{rms} is the current measured by the SR865, this means the contributions of the graphite/BBG interface, the graphite leads, the metallic leads etc. are still present. This makes the attribution of R_{2T} to physics in the BBG/BN doubtful. The resistance of the graphite leads will depend on the applied gate voltage, and its temperature dependence will also contribute to $R_{2T}(T)$. Four-terminal measurements are the standard precisely to exclude these effects. Can the authors provide four-terminal data to eliminate contributions from other sources?

Author's response: We thank the reviewer for raising a valid and important issue. The 2T resistance was calculated as $V_{drive}/I_{rms} - 1M\Omega$ as the reviewer correctly guessed. We used these 2-terminal values in **Fig. 4** (original manuscript) because the measured voltage in the 4-terminal configuration value was saturated at the vertical ($t=0$) features, making the measurements inaccurate. Since we were primarily interested in dissipationless edge states with very small voltage drops, we performed most of our measurements with the sensitivity of the voltage measurement device set to low. On the other hand, the current measurements do not have this saturation issue because the measured current (I_{rms}) only decrease from the initial setting of $V_{drive}/1M\Omega \sim 1nA$. However, as the reviewer correctly pointed out, this 2-terminal measurement includes several unwanted contributions from the BBG/graphite/metal interfaces and graphite, metallic leads itself.

So, we performed additional four-terminal magnetoresistance measurements by setting the voltage measurement range high enough. (**Fig. R2**) The 4-probe measurements confirmed that the feature of decreasing resistance with temperature below 150 mK disappeared, and the gap measured at the same point as before was larger than the original 2 probe measurement value. We replaced **Fig. 4** with the newly measured 4 probe data.

Fig. R2 | Four-probe measurement data with extended voltage measurement range. **a**, Landau fan diagram with the same condition as **Fig. 2a**. Voltage measurement range was set high enough to prevent saturation issues. **b**, Temperature dependence of (0,-2) state at $B=9.6T$.

-The entire manuscript rests on a single device. If at all possible, can the authors present data from a similar device that support the conclusions? At zero degree BBG/BN alignment, making another device does not rely on precise angle control. High-quality stacks should "snap" back to zero degrees alignment upon annealing.

Author's response: We agree with the reviewer on this point and thus measured 3 more samples that supplement our work.

Magnetotransport measurement of one of the new samples that reproduced the valley-selective moiré effect in each Landau level and D field tunability of Chern insulator states is presented below (**Fig. R3**). We added the data and description of magnetotransport measurements of the new samples in the Supplementary section 7 of the revised manuscript.

As a side note, we have attempted to fabricate several new devices. As the reviewer suggested, the graphene sandwiched between hBN tends to snap back to zero degree angle upon annealing, and typically moves a few μm during this process. However, we found that the top graphite sandwiched between the pick up hBN and the top hBN also moves. For many samples, both the graphene and top graphite moved after annealing, often disrupting the intended positions during the stacking process and failing the fabrication. (**Fig. R4**) Therefore to overcome the issue, we sacrificed the exact zero-degree alignment by skipping the annealing process and defining the device geometry first. After that, we were able to fabricate 3 devices with the crystal axis of the BBG and hBN nearly aligned at zero degree.

Fig. R3 | Magnetotransport measurement of 1.37 deg sample. a-c, Landau fan diagram of longitudinal resistance up to $B=14\text{T}$ at different vertical displacement fields. All measurements were conducted at $T=30\text{mK}$. **d-e**, Longitudinal resistance R_{xx} as a function of the carrier density n and D field at 8.9T (**d**), 11.75T (**e**). N denotes the orbital number of the Landau level of BBG and white numbers in **a**, **d** denote the Landau filling factor. Each colored horizontal line in **d-e** corresponds to the same colored lines in **a-c**.

In the supplementary section of the revised manuscript:

“We observed integer quantum Hall, Chern insulator, and fractional quantum Hall (up to $N=3$ LL) states in the Hofstadter butterfly pattern of the new sample, but not symmetry broken Chern insulator and fractional Chern insulator states. For the device of the main text, interacting states appeared in the low (less than $1 \times 10^{12} \text{ cm}^{-2}$) carrier density region, but the new device required about three times higher carrier densities to fill the same superlattice filling factor, so we could observe only the high

density and low ϕ/ϕ_0 region of Hofstadter patterns in the magnetic field range of our system. We believe that this device will also show interaction driven states if the magnetic field is further increased to access the low carrier density region of the Hofstadter pattern.”

Fig. R4 | Stacking configuration of a sample just after the stacking (a) and after annealing (b). In this case, all flakes except the bottom hBN and bottom graphite moved after annealing.

-The thickness of each layer should be mentioned in the Device Fabrication section. Especially the thickness of the BN spacers between BBG and the gates is important, as the screening by the gates can influence the existence of the fragile Chern insulating states.

Author’s response: We added the thickness information of flakes in the Device Fabrication section. The thickness of hBN is approximately 34 nm for top and 61 nm for bottom. We didn't check the exact thickness of graphite gates with AFM but from the optical contrast, each graphite gate flake has more than 7~8 layers.

-The authors mention that "conventional metal contacts can open gaps in BBG" (line 92). A citation for this would be appreciated, and a value of the contact resistance of the graphite contacts at 30 mK and high magnetic field.

Author’s response: We thank the reviewer for the valuable comment. We added references about the interface of metal electrode and bilayer graphene^{6,7} at the line the reviewer mentioned. At the interface of a metal electrode and an underlying BBG, interfacial charge transfer can occur to align the Fermi level of each material, which can introduce an interlayer potential difference between graphene sheets of BBG and open a bandgap.

Also, we conducted an additional 2-probe measurement to estimate the value of contact resistance. At T=30 mK and B ~ 14 T, the contact resistance was measured to be about 20 kΩ. Please also check the full 2-probe magnetoresistance measurement results in **Fig. R9** (a part of the answer to Reviewer #3).

-Figure 1d shows the mobility of the device. Is this the Hall mobility, or the mobility from the onset of SdH oscillations? How relevant is the mobility of a device that is guaranteed to be ballistic at 30 mK in this discussion? The clear Hofstadter butterfly of Figure 2 is enough indication of the cleanliness of this device. In the ultra-quantum limit, the quantum lifetime τ_q is the only useful measure of the quality of a device.

Author's response: First of all, the mobility shown in **Fig. 1d** is the Hall mobility. In the original manuscript, we have tried to give information as much as possible for readers and presented it as a reference for readers to compare with the mobility of graphene or other 2d electron systems presented in the literature.

Yes, we totally agree with the reviewer that the device quality at the ultra-quantum limit should be estimated by τ_q . However, we didn't attempt to estimate τ_q by looking at the onset of SdH oscillations due to the following technical difficulties: We only performed magnetotransport measurements up to 4 K, below which we found that the quantum Hall effect and the moire effect appear at very low magnetic fields, distorting the SdH oscillations and thus making it difficult to estimate τ_q by fitting to the Lifshitz-Kosevich formula.

Given the limitation, in the revised manuscript, we made it clear that the mobility presented in the figure is the Hall mobility, while leaving it to readers to judge how useful that number would be. We appreciate the reviewer's expert comment on this matter.

Reply to reviewer 3's report

Summary of Presented Work

The authors present magnetotransport measurements on a Bernal bilayer graphene device with its crystal axes aligned to one of its encapsulating hexagonal boron nitrides. They observe a rich set of states associated with the moire pattern formed by alignment to the hBN, most of which can be explained by the known Landau Level structure of bilayer graphene and Hofstadter spectrum resulting from the moire pattern and high magnetic field. They analyze in more detail than previous works the effects of displacement field and the longitudinal resistance. Their main experimental claims are:

- 1) The observation of valley-dependent (rather than layer-dependent) moire-induced scattering observed as “bright” R_{xx} features in displacement field vs. carrier density plots.*
- 2) Identification of quantum valley Hall insulators with insulators with $t=0$.*
- 3) Observation of interaction-driven states, specifically FCI and SBCI observed for the first time in R_{xy} plateaux.*

Major comments

Overall, this paper presents the most complete transport picture of hBN-aligned BBG so far. As a result, the cacophony of states make parsing this data quite a task even for experts. I believe that the work requires another pass to make it more digestible to a broader audience, including clarifying the visual presentation of figures, and perhaps presenting more closely zoomed in regions of the data in Fig.5 , for example.

Author's response: We thank the reviewer for acknowledging the importance of our work. The reviewer's valuable comments were very helpful in improving our manuscript. We made every effort to address the points raised by the reviewer. We tried to clarify the text and the visual presentation of the figures, including those mentioned in the minor comments. Here, we present a more closely zoomed in region of the data in **Fig. 5**, now in the **Extended Data Fig. 1f** of revised manuscript, as an example.

Fig. R5 | Zoomed in region of the data in **Fig. 5**

1) *The valley-dependent broadening is an intriguing observation and in my mind well supported by the data.*

Author's response: We are happy to hear the reviewer's positive opinion. Please also see **Fig. R3** for reproducibility on another sample.

2) *I am not convinced that one can confidently identify the $t=0$ features shown in Fig.4 as the quantum valley Hall effect. Claiming the existence of helical edge states based on only a non-monotonic temperature dependence is well below the standard applied to Quantum spin Hall systems both in semiconductor heterostructures and more relevantly at $\nu = 0$ in monolayer graphene. As evidenced by the data itself, there are many competing interacting and non-interacting states in the phase diagram, so I am not sure that one can attribute the non-monotonic dependence to edge states vs. disordered competition between multiple states with different, temperature-dependent resistances.*

For the claims of edge states, some sort of reconfiguration of contacts showing length-dependent transport should be possible to bolster these claims.

Additionally, it would be useful to explain in more detail why one would expect the QVH effect instead of quantum spin Hall, some more complex spin-valley magnetic state, or a charge density wave here would be useful in bolstering the claim of QVH.

Author's response: We appreciate the reviewer for his/her suggestions on the argument of helical edge states at $t=0$ features. We agree that the non-monotonic temperature dependence alone is insufficient to claim the existence of helical edge states. Therefore, we performed non-local transport measurements by changing the contact configuration as suggested by the reviewer.

The left column of **Fig. R6** below shows the longitudinal resistance (R_L) measurement, and the right column shows the non-local resistance (R_{NL}) measurement. When comparing the two data sets, the state (0,-1) is particularly standing out with unusually high values of R_{NL} over the corresponding R_L .

Our interpretation of the data is based on the following analysis: (We believe this is also consistent with the observation of *Sanchez-Yamagishi et al. Nature nanotechnology 12 (2), 118-122 (2017)*)

1. Compressible state: Intermediate R_L , very low R_{NL}
 - Reasoning: The voltage drops for R_{NL} will be decaying far from the source-drain electrodes.
2. Incompressible state without an edge channel: High R_L , very low R_{NL}
 - Reasoning: The voltage drops for R_{NL} will be decaying far from the source-drain electrodes.
3. Incompressible state with chiral edges: Zero R_L , zero R_{NL}
 - Reasoning: Chiral edges would act as equipotential 1D channels to give nearly zero NL resistance
4. Disordered inhomogeneous phase of multiple states: Intermediate R_L , low R_{NL}
 - Reasoning: Inevitable dissipations occur at every boundary of different states and thus make the situation *qualitatively similar* to the one of the compressible states.
5. Incompressible state with helical edges: a sizable fraction of the quantum conductance for both R_L and R_{NL}
 - Reasoning: The voltage drops occur near ohmic contacts due to selective mixing of helical 1D channels.

As expected from the above analysis, in **Fig. R6d**, most compressible states and chiral incompressible states display highly suppressed R_{NL} . Looking at the states of $t = 0$, we can see that they generally have sizable R_{NL} 's. The R_{NL} for (t,s)=(0,-4) is around 1 k Ω at all magnetic field values. The R_{NL} for (0,-2) is similar in size, except for some regions near $\phi/\phi_0 \sim 0.2, 0.4$. For (0,-1), while the longitudinal resistance peak is the smallest of the three, R_{NL} shows the largest value, approaching $h/4e^2$ at high magnetic fields, which is the expected value to be measured for states with a single helical edge when the NL configuration is used as shown in **Fig. R6b**, based on the Landauer-Büttiker formalism. This strongly suggests the existence of non-local helical transport at (0,-1). In the same vein, the state at (0,-2) near $\phi/\phi_0 \sim 0.4$ also hints on a helical state.

For the question about the nature of the state that can harbor a helical edge in this system, we think it is a state having the valley-dependent Chern numbers with opposite signs at K and K', with suppressed intervalley scattering. Actually, our

calculation of the valley-specific Hofstadter spectra in **Fig. S8** shows that, in ZLL, where the observed (0,-1) state resides, the K valley has a Chern number of -2 over a wide range of energy (the absence of states for K) while the K' valley displays moire-induced subbands with many different Chern numbers. From the fact that the total Chern number of the observed (0,-1) state should be 0 means that an energy gap opens at the Fermi level with the Chern number of +2 in the K' valley. We originally named this state as a QVH state, in which the valleys have edge channels with opposite chirality, forming helical edge states.

We note that, since the spectra in **Fig. S8** are limited to the spin degenerate case, a spin hall state or a complex spin-valley magnetic state are not considered and thus cannot be ruled out given the spin degeneracy is to be broken in the real system. Nonetheless, we think that the valley-dependent Chern number provides a crucial mechanism in the formation of the helical edge state.

To sum up, we believe new NL measurements can support that some of the states with $t=0$ have helical edges. At the same time, we admit that our original claim with the quantum Valley Hall effect was a bit aggressive, and thus are willing to tone down the claim as follows:

In the original manuscript:

“Furthermore, the resistance measured for the (0,-2) state in Fig. 4b initially increases with decreasing temperature, but saturates and decreases below 150 mK, signaling the existence of helical edge states of a quantum valley Hall insulating state.”

In the revised manuscript:

“We attribute the high non-local resistance of the state to a helical edge channel of counter-propagating valleys (see Supplementary Information S8). However, we can not rule out the complicated involvement of spins in the edge channel, such as one in a spin-valley magnetic state. A measurement with a large in-plane magnetic field to control Zeeman energy may help to resolve the question while it is out of the scope of the current work.”

In the supplementary information of the revised manuscript:

“For the question about the nature of the state that can harbor a helical edge in this system, we think it is a state having the valley-dependent Chern numbers with opposite signs at K and K', with suppressed intervalley scattering. Actually, our calculation of the valley-specific Hofstadter spectra in **Fig. S8** shows that, in ZLL, where the observed (0,-1) state resides, the K valley has a Chern number of -2 over a wide range of energy (the absence of states for K) while the K' valley displays moire-induced subbands with many different Chern numbers. From the fact that the total Chern number of the observed (0,-1) state should be 0 means that an energy gap opens at the Fermi level with the Chern number of +2 in the K' valley. We originally named this state as a QVH state, in which the valleys have edge channels with opposite chirality, forming helical edge states.

We note that, since the spectra in **Fig. S8** are limited to the spin degenerate case, a spin hall state or a complex spin-valley magnetic state are not considered and thus cannot be ruled out given the spin degeneracy is to be broken in the real system. Nonetheless, we think that the valley-dependent Chern number provides a crucial mechanism in the formation of the helical edge state.”

Fig. R6 | Configuration dependent magnetoresistance measurements. **a,b.** Measurement configuration of longitudinal (**a**, R_{xx}) and non-local (**b**, R_{NL}) resistance. **c,d.** Landau fan diagram of R_{xx} (**c**) and R_{NL} (**d**). **e,f.** Line cuts of R_{xx} (**e**) and R_{NL} (**f**) along the magnetic field direction at $n/n_0 = -1$ (red), -2 (green), and -4 (blue). Dashed line in **f** represents $h/4e^2$.

3) The very nice R_{xy} data and gap measurements are very welcome additions, however I have concerns with the identification of the FCI state, in particular.

The authors identify a single FCI state with fractional t in their data at $(-14/3, 5/9)$, located in a $C=3$ band originating at $1/3$ of a flux quantum per moire unit cell. I do not believe the manuscript provides enough evidence that this state is not in fact another nearby SBCI state $(-5, -2/3)$. I'll outline why I think this is the case below:

a) The plateau at $R_{xy}=3/14 h/e^2$ is not convincing. If the authors would like to make this claim, the plateau should be more clearly presented. The R_{xx} minimum of this state is the least developed of the R_{xx} dips, as one would expect from the very small measured gap. Strictly speaking, this means that the plateau should not be quantized, and based on what I can see could just as well be a poorly developed $1/5 h/e^2$ plateau. I think a better presentation and discussion of the plateau is warranted in either case.

b) In the absence of more clear data, I think that while an FCI is possible at these t and s values, it is much more likely that it is energetically favorable for the $C=3$ band to breaking into three, symmetry broken $C=1$ bands due to interactions, than an FCI appearing in an already symmetry broken $C=2$ band.

A back of the envelope way of estimating the strength of interactions in the Hofstadter spectrum is to look at the single-particle bands 'width' in carrier density at a given field, which we can take as the overall degeneracy of the band (or similarly we can relate it to an effective magnetic length which governs Coulomb interactions). The state in question appears while the triangle of the $C=3$ band is still quite narrow, much narrower than the LL at 5 Tesla in which the traditional FQH state first appears. So, for an FCI to appear in the regions indicated by Fig.5, it would have to somehow be more robust than the highest energy gap FQH state in the system, which seems very unlikely to me given the finite bandwidth associated with the Hofstadter butterfly. I think a positive identification of this state would require a lot of theoretical explanation beyond what is provided.

Author's response: The reviewer raised a valid and important issue about the identification of FCI states. To check the R_{xy} plateau in the state we claimed to be a FCI, in **Fig. R7**, we performed additional R_{xy} measurements with a 10 times finer variation of the gating voltage compared to our previous data. After a careful assessment, we now agree that the state we claimed to be $(-14/3, 5/9)$ FCI was actually nearby $(-5, 2/3)$ SBCI, whose gap was so small that the $h/5e^2$ plateau was not well formed. Therefore, we modified **Fig. 5** and stated the confirmation of a SBCI rather than a FCI. We believe that from the quality of the device, FCIs can be found at higher magnetic fields.

The reason why we wrongly identified and how we corrected is as follows:

In our previous data, we estimated the plateau from R_{xy} values at magnetic fields around 9.5T to 9.6T, and from additional measurements of **Fig. R7b** we learned that there is a subtle magnetic field dependence of the measured quantizations. While not knowing the exact reason for this, we conclude that this is what caused us to slightly overestimate the value of R_{xy} .

For typical $t \neq 0$ incompressible features, estimating the Hall resistance plateau value using $(R_{xy}(n, B) - R_{xy}(n, -B))/2$ is fairly accurate because R_{xx} itself goes to zero and the plateau has some width, but for weak features that appear in a very narrow range and R_{xx} doesn't go to zero completely, we think that even a small offset in carrier density may have caused the error in the estimation of the plateau value.

We revised the manuscript to reflect this change, as follows:

In the original manuscript:

“We interpret that the observed FCI at $(-14/3, 5/9) = (v, 0) + \nu_C (\Delta t, \Delta s) = (-3, 0) - 5/9 \times (3, -1)$ fills the Chern band of Chern index $\Delta t = +3$, and the superlattice filling index $\Delta s = 1$ with the filling $\nu_C = 5/9$ just below the $\nu = -3$ IQHE gap, which is thought to break sublattice symmetry and hold carriers of the fundamental charge of $e/3$ ”

In the revised manuscript:

“These observed interaction driven states have gap trajectories described by $(t, s) = (t_L, s_L) + \nu_C (\Delta t = t_R - t_L, \Delta s = s_R - s_L)$ (where $(t_{L,R}, s_{L,R})$ is the neighboring left and right integer states respectively) with a fractional ν_C . We observed states with $\nu_C = 1/2$ in bands with Chern number $\Delta t = -2$ $((-5, 1/2), (-3, 1/2))$ and states with $\nu_C = 1/3$ and $2/3$ in a band with $\Delta t = 3$ $((-5, 2/3), (-4, 1/3))$.”

Fig. R7 | Hall measurement data at $T=30\text{mK}$, $D=91\text{mV nm}^{-1}$ **a**, Line cuts of Landau fan diagram of $(R_{xy}(n, B) - R_{xy}(n, -B))/2$ along the density axis. Each line was measured by varying the magnetic field from 9.65T to 9.85T in 0.05T steps. **b**, A zoomed-in view of the region indicated by the red dashed box in **a**.

Summary

To sum up, I think that the data and much of the discussion here is very intriguing to experts in the subfield, however there are major outstanding issues I would need to see addressed to recommend publication on Nature Communications. Please also see some more minor comments and corrections provided below.

In terms of significance I believe this work is original and of interest to many people working on graphene and other 2D materials.

Author's response: We thank the reviewer again for acknowledging the importance of our work. We also greatly appreciate the reviewer's very detailed and helpful comments.

Minor comments:

Around line 98, it is stated that energy gaps are expected, but only very small peaks in R_{xx} are observed. A brief comment on this would be helpful to the reader.

Author's response: We described the satellite peaks as insulators based on the references and our own continuum calculation.^{8,9} In Ref (8), the gap opens at the corner of the Moire Brillouin zone, and in Ref (9), it was shown that the gap can be open or closed depending on the parameters used to calculate the band structure. In a recent paper¹⁰ on a slightly (0.75 deg) twisted BBG-hBN system, band structure fitting of magnetization measurement data shows that there is not a full gap at $n/n_0 = \pm 4$, but a DOS minimum, which is manifested as small peaks of ρ_{xx} in transport measurements.

For nonzero magnetic field, $n/n_0 = -4$ peak showed insulating behavior. (**Fig. R8b**) But at zero magnetic field, it was difficult to confirm whether the $n/n_0 = -4$ peak is an insulator up to 4K (**Fig. R8a**). So, we conducted additional transport measurements at higher temperatures to check whether the satellite peak is an insulator. **Fig. R8c** shows that the satellite peak becomes slightly larger when the temperature is increased by a few tens of K, and the resistance value continues to increase as the temperature is increased to 300 K. Therefore, we concluded that it is more appropriate to interpret the $n/n_0 = -4$ peak as coming from a DOS minimum, rather than a full gap.

Thus, to avoid potentially misleading information on the existence of the full gap in the original manuscript, we have carefully revised the statements.

In the original manuscript:

“induced energy gaps”

In the revised manuscript:

“induced energy gap or DOS minimum”

“Actually, a recent experiment supports DOS minimum rather than full gap.” (added)

Fig. R8 | Temperature dependent longitudinal resistance measurement at $D=0$. **a**, R_{xx} peak of $n/n_0 = -4$ at $B=0T$ up to $T=4K$. **b**, R_{xx} peak of $n/n_0 = -4$ at $B=0.2T$ up to $T=4K$. **c**, High temperature R_{xx} measurements at $B=0$ up to $T=300K$

Fig 2b is very busy, specifically it is very hard to distinguish the QH gray lines and the Diophantine gray lines. Personally, I also think that the Brown Zak lines could be removed or made less intrusive.

For all of the resistance figures, it should be stated or otherwise indicated on the colormap that the color scale is truncated at 10, 15, etc. kOhms, as there are much more resistive features washed out by the choice of color limits (necessarily).

Fig.3d,e. The thick dashed lines are confusing as the gaps between dashes are very thick and at first glance appeared to be marking places where features didn't exist.

Between Fig5b and C the colors of arrows for the different states perhaps should match.

Author's response: We appreciate the reviewer's detailed comments to improve readability of the paper. We have addressed each of the points by modifying the figure and adding additional comments in the main text as suggested

1. We have lightened the color of the Diophantine lines in **Fig. 2b** and changed the Brown Zak lines to thinner dashed lines.
2. As the reviewer correctly pointed out, many resistive features, especially $t=0$ features, were washed out by our choice of colormap. So, we clarified in the main text that each colorbar was truncated at the end value.
3. We adjusted the gap and thickness of the dashed lines in **Fig. 3d,e**.
4. Since we have corrected the $(-14/3, 5/9)$ FCI to $(-5, 2/3)$ SBCI in the previous response, we have matched the color of the arrows to red.

I'd like to see any characterization of the graphite contacts, as from personal experience and discussions with others, I would expect graphite contacts to be uniquely bad for transport at high fields, especially if the crystal axis of the graphene and graphite are not very close.

Author's response: We appreciate the reviewer's expert comment. Indeed, we have sometimes observed poor contact in high magnetic fields when using graphite contacts in other devices. We think it is possible that momentum mismatch due to crystal axis angle differences can cause poor contact at the graphene/graphite vdW interface. Actually, we did not pay special attention to the angle between the contact graphite and BBG during the stacking process, but upon checking, the device in the main text seems to be stacked with the angle between BBG/graphite flakes more or less parallel, and it is possible that they were further aligned during the annealing process.

To characterize the contact resistance, we conducted an additional 2-probe magnetoresistance measurement at $T = 30$ mK.

At zero field, the lowest value of the two-probe resistance was measured to be about 8 k Ω , which directly gives an approximate contact resistance due to the small 4-probe resistance. At high magnetic fields, we compared the resistance values of the $\nu=-4$ and $\nu=-8$ IQHEs, which are among the strongest features in the single particle picture, and found them to be about 28.7 k Ω and 25.5 k Ω , respectively. We analyzed that the difference, about 3.2 k Ω , came from the difference in conductance of the edge states ($(h/e^2)/8 \sim 3.2$ k Ω), so we estimated the residual resistance to be about 22 k Ω . Subtracting 2 k Ω of cryostat wire resistance from this, the contact resistance is approximately 6 k Ω at zero field and 20 k Ω at 12.35 T. (This is not much different at 14 T) Although it increases by about a factor of 3, we found the graphite contact remained good enough that even the two-probe measurement was able to reproduce most of the detailed features observed in the four-probe measurement.

We added this characterization of the contacts to the supplementary information in the revised manuscript.

Fig. R9 | Contact resistance characterization. **a**, Measurement configuration of 2 probe resistance. Two legs are paired in parallel and the series resistance of the two pairs was measured. This makes the measured quantity approximate the resistance coming from a single leg. **b**, Landau fan diagram of 2-probe resistance at $T = 30$ mK, $D = 116$ mV/nm $^{-1}$. The colormap is truncated at both ends. **c**, Line cut at $B = 12.35$ T ($\phi/\phi_0 = 0.5$, white dashed line in **b**). The red arrows in **b,c** indicate Landau filling factor -4 and -8 points. **d**, Line cut at $B = 0$ T. The 8 k Ω line is shown as a black dashed line. The black arrows denote the area affected by the partially gated legs.

References

1. Barlas, Y., Côté, R., Nomura, K. & MacDonald, A. H. Intra-Landau-Level Cyclotron Resonance in Bilayer Graphene. *Phys. Rev. Lett.* **101**, 097601 (2008).
2. Spanton, E. M. *et al.* Observation of fractional Chern insulators in a van der Waals heterostructure. *Science* **360**, 62–66 (2018).
3. Hunt, B. M. *et al.* Direct measurement of discrete valley and orbital quantum numbers in bilayer graphene. *Nat Commun* **8**, 948 (2017).
4. Liu, Z., Bergholtz, E. J., Fan, H. & Läuchli, A. M. Fractional Chern Insulators in Topological Flat Bands with Higher Chern Number. *Phys. Rev. Lett.* **109**, 186805 (2012).
5. Möller, G. & Cooper, N. R. Fractional Chern Insulators in Harper-Hofstadter Bands with Higher Chern Number. *Phys. Rev. Lett.* **115**, 126401 (2015).
6. Nouchi, R. Experimental signature of bandgap opening in bilayer graphene at metal contacts. *Applied Physics Letters* **105**, 223106 (2014).
7. Zheng, J. *et al.* Interfacial Properties of Bilayer and Trilayer Graphene on Metal Substrates. *Sci Rep* **3**, 2081 (2013).
8. Moon, P. & Koshino, M. Electronic properties of graphene/hexagonal-boron-nitride moiré superlattice. *Phys. Rev. B* **90**, 155406 (2014).
9. Chen, X., Wallbank, J. R., Mucha-Kruczyński, M., McCann, E. & Fal'ko, V. I. Zero-energy modes and valley asymmetry in the Hofstadter spectrum of bilayer graphene van der Waals heterostructures with hBN. *Phys. Rev. B* **94**, 045442 (2016).
10. Bocarsly, M. *et al.* De Haas–van Alphen spectroscopy and magnetic breakdown in moiré graphene. *Science* **383**, 42–48 (2024).

REVIEWERS' COMMENTS

Reviewer #1 (Remarks to the Author):

The authors have addressed my comments and replied satisfactorily. In my opinion, after the revision the manuscript is improved.

Reviewer #2 (Remarks to the Author):

The authors have satisfactorily addressed my comments and I recommend this manuscript for publication. In particular, I commend the authors for fabricating and measuring three more devices.

As an aside, an old experimental trick to deal with voltages spanning many orders of magnitude is to connect two voltmeters at different sensitivities to the same pair of potential probes. When the meter at small sensitivity saturates, the other one still records the voltage accurately (at the cost of using two instruments).

Reviewer #3 (Remarks to the Author):

In my opinion, The authors have meticulously responded to all referee comments and made great positive changes to the work. In its current state I recommend the manuscript for publication in Nature Communications.

Reply to reviewer 1's report

“The authors have addressed my comments and replied satisfactorily. In my opinion, after the revision the manuscript is improved.”

Author's response: We thank for the reviewer's positive response to our reply. The reviewer's expert comments gave us a lot to think about and helped us improve the manuscript.

Reply to reviewer 2's report

The authors have satisfactorily addressed my comments and I recommend this manuscript for publication. In particular, I commend the authors for fabricating and measuring three more devices.

As an aside, an old experimental trick to deal with voltages spanning many orders of magnitude is to connect two voltmeters at different sensitivities to the same pair of potential probes. When the meter at small sensitivity saturates, the other one still records the voltage accurately (at the cost of using two instruments).

Author's response: We thank for the reviewer's positive response to our reply. The addition of the measurement data from the newly fabricated device made our manuscript more concrete, and the trick that the reviewer pointed out will be very useful for our next experiment.

Reply to reviewer 3's report

In my opinion, The authors have meticulously responded to all referee comments and made great positive changes to the work. In its current state I recommend the manuscript for publication in Nature Communications.

Author's response: We thank for the reviewer's positive response to our reply. The reviewer analyzed our data very thoroughly, which helped us to improve our manuscript in the process of responding.